# Liver Injury Associated with COVID-19 Infection: Pathogenesis, Histopathology, Prognosis, and Treatment

**DOI:** 10.3390/jcm12052067

**Published:** 2023-03-06

**Authors:** Noha Mousaad Elemam, Iman M. Talaat, Azzam A. Maghazachi, Maha Saber-Ayad

**Affiliations:** 1College of Medicine, University of Sharjah, Sharjah 27272, United Arab Emirates; 2Sharjah Institute for Medical Research, University of Sharjah, Sharjah 27272, United Arab Emirates; 3Faculty of Medicine, Alexandria University, Alexandria 21131, Egypt; 4Faculty of Medicine, Cairo University, Cairo 11956, Egypt

**Keywords:** NAFLD, HCV, HBV, COVID-19, cytokine storm, drug-induced liver injury

## Abstract

Liver injury occurs frequently as a consequence of SARS-CoV-2 infection. Direct infection of the liver leads to hepatic impairment with elevated transaminases. In addition, severe COVID-19 is characterized by cytokine release syndrome, which may initiate or exacerbate liver injury. In patients with cirrhosis, SARS-CoV-2 infection is associated with acute-on-chronic liver failure. The Middle East and North Africa (MENA) region is one of the world’s regions characterized by a high prevalence of chronic liver diseases. Both parenchymal and vascular types of injury contribute to liver failure in COVID-19, with a myriad of pro-inflammatory cytokines playing a major role in perpetuating liver injury. Additionally, hypoxia and coagulopathy complicate such a condition. This review discusses the risk factors, and the underlying causes of impaired liver functions in COVID-19, with a focus on key players in the pathogenesis of liver injury. It also highlights the histopathological changes encountered in postmortem liver tissues as well as potential predictors and prognostic factors of such injury, in addition to the management strategies to ameliorate liver damage.

## 1. Introduction

In March 2020, the coronavirus disease-2019 (COVID-19) was announced as a global pandemic due to the outbreak of the novel virus, SARS-CoV-2 that originated in Wuhan, China in late 2019 [1]. Like many other beta group members of the Coronaviridae family, the transmission of this virus was found to be through respiratory droplets and aerosols [2,3]. COVID-19 patients could have varying degrees of symptoms ranging from asymptomatic to severe/critical conditions [4]. Moreover, COVID-19 could cause respiratory complications such as hypoxemia, dyspnea, and acute respiratory distress syndrome (ARDS) [5]. Besides that, COVID-19 was reported to affect the function of other organs and play a role in tissue injury and extra-pulmonary manifestations. For example, COVID-19 was associated with damage to the hematological, liver, kidney, cardiac, and gastrointestinal systems, with a high rate of mortality and multi-organ failure in severely infected patients [6]. Previous studies found that abnormal levels of the liver enzymes, alanine aminotransferase (ALT) and aspartate aminotransferase (AST), were frequently encountered in COVID-19 patients, indicating liver damage [7,8,9,10]. In addition, severe and ICU-admitted COVID-19 patients presented with liver dysfunction, as evidenced by high liver enzyme levels [8,11,12,13,14]. On the other hand, pre-existing comorbid conditions, such as liver disease, could exacerbate and have a more severe form of COVID-19 infection [15]. In this review, we aim to highlight the relation between liver injury as a consequence or a pre-existing condition in COVID-19 infection.

## 2. Global Burden of Liver Diseases

According to the published data from global burden of diseases, over 2 million people globally pass away from liver disease each year, with 1 million of those fatalities coming from cirrhosis complications, 1 million from viral hepatitis, and 1 million from hepatocellular carcinoma [16]. Together, liver cancer and cirrhosis account for 3.5% of all fatalities worldwide and are the 11th and 16th most common causes of death worldwide, respectively. Cirrhosis is one of the top 20 causes of disability-adjusted life years (DALY) and years of life lost. Over 2 billion people use alcohol globally, and more than 75 million have been diagnosed with alcohol use disorders or are at risk of developing such health problems. Over a span of 30 years, cirrhosis and other chronic liver diseases witnessed a rise from 1.34% to 1.82% of total DALY, in 1990 and 2019, respectively [17].

The Middle East and North Africa (MENA), in addition to Latin America and the Caribbean witnessed the highest percentage of regional deaths from liver disease. Some countries with the greatest fatality rates from cirrhosis are Egypt, Moldova, and Mongolia. India accounts for one-fifth (18.3%), whereas China accounts for 11% of all cirrhosis fatalities globally due to the burden of the population [18].

The global burden of hepatocellular carcinoma (HCC) and cirrhosis grew rapidly during the last decades. Almost 2 billion adults are obese or overweight, and more than 400 million have diabetes, both of which are risk factors for hepatocellular carcinoma and non-alcoholic fatty liver disease. The incidence of viral hepatitis is still widespread over the world, and drug-induced liver damage is becoming a more significant cause of acute hepatitis.

The prevalence of non-alcoholic fatty liver disease (NAFLD) was found to be growing, causing liver mortality and morbidity [19]. In 2017, there was an estimated 1.5 billion cases of chronic liver disease worldwide, with liver cirrhosis accounting for an estimated 1.32 million deaths [2,20]. Liver cirrhosis could be caused by chronic infection with hepatitis B virus (HBV), hepatitis C virus (HCV), alcohol-related liver disease, and non-alcoholic steatohepatitis (NASH), that are independent risk factors for HCC and cholangiocarcinoma. The global HCV prevalence was estimated at 2.5%, ranging from 2.9% in Africa and 1.3% in America [21].

The most common HCV genotype is genotype 1 (49.1%), followed by genotype 3 (17.9%), genotype 4 (16.8%), genotype 2 (11.0%), and the rare genotypes 5 and 6 (<5%). The majority of HCV-infected patients with genotypes 4 and 5 are found in lower-income nations, while genotypes 1 and 3 are widespread globally. Previously, the highest HCV burden was reported in Asian countries such as China, Pakistan, and India, as well as MENA countries such as Egypt, in addition to the United States [16,22]. The treatment and achievement of a sustained virological response (SVR) were associated with a decrease in HCC development rate. In 2017, around 700,000 HCV cases achieved SVR, with the majority of patients who initiated therapy from the Mediterranean region, particularly Egypt [23,24]. NAFLD embraces 2 conditions: (A) NAFLD including steatosis with/without mild lobular inflammation, and (B) NASH with various degrees of fibrosis, cirrhosis, and HCC [25,26]. There is an increase in NAFLD prevalence that could be attributed to the increase in the presence of co-existing risk factors such as obesity and diabetes [25]. The global prevalence of NAFLD was reported to be around 25% with a higher prevalence (>30%) present in the Middle East [27,28].

According to GLOBOCAN 2020, a high prevalence of HCC was reported with an estimation of 905,677 cases with particularly high rates in East Asia, and North Africa [29,30]. The causes of cirrhosis vary depending on the explored region. For example, in Western and industrialized countries, alcohol and NAFLD are identified as primary causes, whereas in China and other Asian countries, hepatitis B continues to be the major cause of liver cirrhosis [31]. In North African countries, viral hepatitis B and/or C are the major contributing factors to liver cirrhosis [32]. It was previously reported that the highest percentage of death due to liver disease was seen in Latin America as well as MENA region [16]. For instance, Egypt has been reported to have one of the highest cirrhosis mortality rates in the world [33]. In 2020, Egypt was identified to have the second highest incidence and mortality rates for HCC [29].

According to the latest report by Zhai et al., the global age-standardized mortality rate of liver cirrhosis due to viral hepatitis has decreased due to the HBV vaccination and availability of direct-acting antiviral therapy, while rates due to alcohol and NASH have increased [34]. Thus, in order to further reduce the burden of liver diseases, early diagnosis of chronic liver diseases is needed to identify viral hepatitis, steatosis, and liver fibrosis before the development of cirrhosis or HCC. Although liver transplantation is the second most frequent solid organ transplant, current rates only meet around 10% of the world’s transplantation needs. The aforementioned figures are alarming, but they also point to a significant opportunity to advance public health as the majority of liver disease causes are curable.

## 3. Pathogenesis of Liver Injury in COVID-19 Infection

SARS-CoV-2 has widespread organotropism; the liver and numerous other extra-pulmonary tissues have been found to express SARS-CoV-2 RNA [35,36]. Hepatic dysfunction, indicated by liver enzyme abnormalities, is frequently noted in 14–53% of COVID-19 patients, particularly in severe cases, according to various studies [37]. Patients with severe COVID-19 infection have significantly more abnormal liver tests, which are linked to poor outcomes [38]. Of note, there are two categories of liver involvement: those with modest liver test abnormalities, which are typically temporary, and those with greater levels of the liver enzymes [39]. In some studies, severe COVID-19 infection was accompanied by predominately elevated serum transferase levels [38,40]. 

Patients with COVID-19 infection have experienced hepatic damage, especially those with moderate to severe illness. There is currently no known explanation for the pathological alterations generated by SARS-CoV-2 in liver tissue. Furthermore, the mechanisms causing liver damage in the 2019 coronavirus infection have not yet been identified [38,41]. However, there are many theories that might explain abnormal liver dysfunction: 1. SARS-CoV-2-induced direct liver injury, 2. drug-induced liver injury, 3. hypoxia, 4. immunological dysregulation, and 5. cytokine storms (Figure 1). Additionally, underlying liver diseases can worsen liver damage caused by COVID-19 [42].

### 3.1. Direct Cytopathic Damage by the Virus

SARS-CoV-2 can directly bind to the angiotensin-converting enzyme 2 receptor (ACE2), which is expressed in the liver (2.6%) and bile duct (59.7%), to cause liver cell damage [43]. Although Coronavirus particles were described in the cytoplasm of hepatocytes in some studies [37,38,43,44], others did not report viral inclusions in the liver tissue [45]. Other cytopathic histopathological findings, such as apoptosis of liver cells, bi-nucleated cells, swollen mitochondria, and canalicular alteration, also point to a direct viral cytopathic action [37,38].

### 3.2. Drug Hepatotoxicity

The liver, which serves as the body’s primary detoxifier, is impacted by several medications used to treat COVID-19 [37]. Antipyretics, antiviral medications, antibacterial medications, herbal remedies, immunosuppressive agents, and other medications that patients attempt throughout clinical treatment can directly or indirectly result in drug-induced liver impairment [42]. Anti-HIV viral protease inhibitor lopinavir/ritonavir has demonstrated in vitro anti-SARS-CoV-2 action; however, it has been established as an independent risk factor for severe liver injury during pharmacological therapy against SARS-CoV-2 [46,47,48]. Following antiviral therapy, some studies found elevations in liver enzymes [37]. However, other models did not find this link to be significant [38]. It is yet unclear how drug-induced liver damage affects the progress of COVID-19.

### 3.3. Severe Inflammatory Response

As an infectious condition that triggers immune-mediated inflammatory injury, COVID-19 can lead to multi-organ dysfunction in addition to pulmonary inflammation. Patients with severe COVID-19 infection may also have systemic inflammatory response syndrome and hepatic damage [37]. Compared to patients with normal hepatic function, those with liver impairment tend to have a higher incidence of cytokine storm [49]. The sera of SARS-CoV-2-infected patients showed elevated levels of monocyte chemoattractant protein 1, interferon-inducible protein-10, IL-2, IL-6, IL-8, IL-10, and IL-17 [7,50]. 

### 3.4. Other Causes of Hepatic Damage

In severely infected COVID-19 patients, hypoxia, acute respiratory distress syndrome (ARDS), and cardiac failure might predispose to hepatic ischemia, which leads to liver cell inflammation and necrosis [45]. Additionally, sinusoids and vascular thrombosis help explain how coagulation malfunction contributes to liver injury [49]. Patients with COVID-19 may be more vulnerable to subsequent damage because of their pre-existing chronic hepatitis [43]. The effect of antecedent hepatic conditions on COVID-19 infection is still unknown.

### 3.5. Morphological Hepatic Changes

Numerous COVID-19-related autopsy series have been researched all over the world in an effort to identify any potential unique histopathological pattern connected to the viral infection [51,52,53,54,55,56,57]. The dysregulation of circulating liver-associated enzymes, which is observed to be mild to moderate in a considerable number of COVID-19 patients, has been identified as one hepatic presentation of SARS-CoV-2 [58,59,60,61]. Nevertheless, little is known about the hepatic histological characteristics during SARS-CoV-2 infection.

There are inadequate morphological investigations to evaluate the pathological liver abnormalities brought on by COVID-19 infection, and those tend to be post-mortem autopsies. The initial post-mortem autopsy specimen was carried out by Xu et al. [40]. A 50-year-old man died from severe COVID-19 following ARDS. The liver histology revealed mild lobular, portal inflammatory infiltrates, as well as moderate micro-vesicular steatosis. Both drug-induced liver injury and SARS-CoV-2 infection are potential causes of the damage. Additionally, CD4 and CD8 lymphocytes in peripheral blood were noticeably diminished, but extremely reactive with increased cytotoxic granulations in CD8^+^ T cells and pro-inflammatory CCR6+Th17 CD4^+^ cell populations. A study conducted by Liu et al. [62] revealed several hepatic lesions in a COVID-19 patient who was referred to the hospital for repeated cerebral infarctions. The hepatic lesions in this patient included sinusoidal congestion with micro-thrombosis, lymphocytic infiltration, and focal lobular necrosis. Additionally, autopsies performed by Tian et al. [43] for four patients who died of COVID-19 pneumonia revealed mild centrilobular sinusoidal dilatation in the hepatic sections as a non-specific finding. A minimal lobular lymphocytic infiltrate was observed in one postmortem autopsy, while focal macro-vesicular steatosis was observed in another one. Yet, patchy necrosis in the centrilobular and periportal areas was observed in one case. Regenerative nodules and thick fibrous bands were visible in the liver tissue of one patient, which was compatible with his history of cirrhosis. Interestingly, the liver tissue of one of the four cases was used to isolate SARS-CoV-2 RNA using reverse transcriptase-polymerase chain reaction (RT-PCR).

Ji et al. [39] investigated liver dysfunction in 202 patients who were COVID-19 positive. They contrasted the clinical outcomes between patients with and without a history of NAFLD. NAFLD patients exhibited a longer viral shedding time and a higher probability of illness progression. One post-mortem hepatic biopsy revealed micro-vesicular steatosis and excessive T cell activation, which implied liver damage from virally induced cytotoxic T cells.

A study of 48 post-mortem liver biopsies from COVID-19-positive patients was published by Sonzogni et al. [49]. During hospitalization, no substantial clinical complaints of liver disease were observed. Numerous vascular changes were found in the liver samples, including luminal dilation, thrombosis and thickening of the vascular wall due to fibrosis. Yet, findings related to inflammation were minimal. These results imply that endothelial injury or coagulation malfunction might be the primary etiology of COVID-19 liver damage. In a study published by Wang et al. [38], 64 out of 156 COVID-19 patients (41%) had high aminotransferases and presented with more severe illness and higher CO-RADS (COVID-19 Reporting and Data System) scoring. Out of the 156 patients, four passed away and three had abnormal liver enzymes. The pathological findings revealed considerable steatosis and extensive hepatic apoptosis. Other histopathological findings included prominent lymphocytic infiltration within the portal tracts and mild to moderate lobular inflammation. To explore the cytopathic effect caused by SARS-CoV-2 infection, liver tissues were examined by transmission electron microscope. Coronavirus particles were seen inside the hepatocytes. The virions had a spherical form, ranged in size from 60 to 120 nm, resembled spikes, and lacked membrane-bound vesicles. Other cytopathic effects included swollen mitochondria and deposition of dense materials intra-cytoplasmic.

Furthermore, Wang et al. [45] reported different pathological findings from the liver tissue of three patients who died due to COVID-19 infection, including canalicular cholestasis, peri-venular inflammatory cell infiltration, micro-vesicular steatosis, coagulative necrosis in centrilobular regions necrosis and minimal apoptotic liver cells. One of these three patients had a moderate increase in serum aminotransferase. Hepatocytes and cholangiocytes did not contain SARS-CoV-2 RNAs, according to in-situ hybridization. These data suggest that hepatic ischemia or drug-induced liver injury may be the underlying causes of hepatocyte necrosis and lobular hepatitis.

Seven COVID-19 autopsies were reported by Rapkiewciz and coworkers [63]. The most common finding across all organs was thrombosis. In all cases, macro-vesicular steatosis without obvious inflammation was seen in liver samples, as well as platelet-fibrin micro-thrombi within the hepatic sinusoids. A hepatic vein thrombus was noted in one case. Hence, they concluded that thrombosis might contribute to the pathophysiology of COVID-19. In addition, Wichmann et al. [44] performed postmortem biopsies on 12 individuals who died of COVID-19. Hepatomegaly, fatty change and chronic congestion were all visible during the liver examination. Interestingly, SARS-CoV-2 RNA was found in the lungs of all patients by RT-PCR, while five patients had high viral RNA titers in their liver, kidney, and heart.

## 4. Risk Factors and Prognosis of Liver Injury in COVID-19

Male gender and C-reactive protein (CRP) levels were identified as independent risk variables for COVID-19 infection associated with liver damage. Moreover, the mortality of COVID-19 patients with liver injury was significantly influenced by elevated CRP and monocyte counts, as well as reduced lymphocyte counts [64]. Another study by Zhang H. et al. reported that high D-dimer levels, male sex, and high neutrophil percentages were predictive risk factors for liver injury in COVID-19 patients [65]. This was further supported by other studies that indicated male gender, elevated CRP levels, and high neutrophil-to-lymphocyte ratio to be risk factors for COVID-19-associated liver injury [4,66]. Furthermore, another study by Deng H et al. revealed that the male sex and high viral loads in the nasopharynx were independent risk factors for liver damage in COVID-19 patients [67]. Such differences in disease severity due to sex were attributed to the estrogen levels that could be protective in females [68,69]. 

Several studies aimed at exploring potential risk factors for severe liver injury in COVID-19 patients. High cytokine levels such as IL-6 and IL-10 as well as reduced CD4^+^ T lymphocytes were reported to be potential risk factors in patients with COVID-19 developing severe liver injury [45,48]. Additionally, the use of the anti-viral drug, ritonavir, was found to be a risk factor for liver injury in COVID-19 [47]. This indicates that the progression of liver injury was linked to drug use, T lymphocytes, and inflammatory cytokines. A similar association was previously reported by our group where we identified elevated serum levels of IL-6 and granzyme B to predict liver injury in COVID-19 patients [70]. It is noteworthy to mention that a study by Wang M. et al. identified COVID-19 patients with liver injury to have high serum levels of ferritin, high-sensitivity CRP, procalcitonin, interleukin-2 receptor (IL-2R), tumor necrosis factor-α (TNF-α), erythrocyte sedimentation rate, gamma-glutamyl transferase (γ-GT), and lactate dehydrogenase (LDH) [64,66]. This suggests that SARS-CoV-2 causes liver impairment by targeting hepatocytes and cholangiocytes. A multicentered study by Yu D. et al. described abnormal AST levels to be the most associated indicator with the mortality risk in hospitalized COVID-19 patients [4,71]. Another study confirmed these findings where elevated AST levels and AST/ALT ratio were described as possible indicators for the COVID-19 progression, poor prognosis, and high mortality that could be a consequence of liver cirrhosis [72]. Multiple studies suggested that liver injury influenced the prognosis of COVID-19 patients, as the severity and mortality rates were higher in those patients with liver injury [4,73,74].

The effect of pre-existing liver diseases on COVID-19-associated liver injury showed controversial results in the literature. COVID-19 patients with NAFLD were reported to have abnormal liver function and were at a higher risk of disease progression [39]. In addition, cirrhotic COVID-19 patients were found to be at a higher risk of hospitalization and mortality [75]. Moreover, a study by Chen F. et al. identified hepatic steatosis as an independent risk factor for liver injury in COVID-19 infection [76]. Additionally, patients suffering from viral hepatitis (hepatitis B and C) developed more severe liver dysfunction upon encountering COVID-19 infection, which could be attributed to the pre-existing immune dysregulation [66]. On the contrary, studies showed that patients with pre-existing liver disease or hepatitis B infection did not show more severe symptoms of COVID-19 disease, thus viral hepatitis cannot be considered as an independent risk factor for liver injury [11,38,77,78].

## 5. Treatment Strategies for Liver Injury in Patients with COVID-19

The incidence of liver injury associated with COVID-19 was found to be inconsistent in the literature. This could be attributed to the variability of liver injury definition and evaluation parameters, and the inconsistent statistical time points [79]. Therefore, management strategies were variable in different countries. Patients with COVID-19 presenting with liver damage could be treated with hepatoprotective, anti-jaundice, or anti-inflammatory drugs, such as polyene phosphatidylcholine (PPC), glycyrrhizic acid, Urso deoxycholic acid, and adenosylmethionine. Previously, PPC has shown efficacy in specific liver diseases [80]. However, the drug bicyclol was found to be superior in terms of ameliorating liver enzymes [81]. Till the moment, there is no conclusive preference for either in the case of COVID-19-related liver injury.

In critically ill patients infected with SARS-CoV-2, a limited number of medications should be chosen to avoid drug misuse and aggravation of liver burden and to reduce drug-drug interactions. Interestingly, a study by Hoever et al., revealed that glycyrrhizic acid derivatives may also have antiviral activity against SARS-CoV-2 [82]. In addition, glycyrrhizic acid has a strong affinity for liver enzymes responsible for steroid metabolism and hinders the inactivation of cortisol and aldosterone. It also shows obvious corticosteroid-like effects, such as anti-inflammatory, anti-allergic, and protective film structures, without obvious cortical hormone-like side effects [82]. Interestingly, glycyrrhizic acid is one of the major active compounds in licorice and has a potential immunomodulatory effect [83,84].

Suppressing the inflammatory response and minimizing hypoxia are key components in managing liver injury related to COVID-19, as reported by Hu et al. [85]. In their study, they explored the clinical characteristics, susceptible population, and treatment strategies for patients with SARS-CoV-2 infection and concluded the importance of those two factors in addition to the supportive treatment. Conservative oxygen therapy is preferred, and ventilator-associated pneumonia should be strictly supervised in patients receiving mechanical ventilation [86]. Xu et al. showed that an artificial liver blood purification system could improve the treatment effect in critically ill patients by rapidly removing inflammatory mediators, blocking cytokine storms, and improving the water-electrolyte balance [86]. Similar findings were reported by Liu et al. who showed a significant declining trend in the levels of cytokines and inflammatory factors (IL-6 and CRP) in patients with COVID-19 after implementing the artificial liver blood purification [87].

Patients receiving mycophenolate (immunosuppressant therapy for liver transplantation) are more prone to develop severe COVID-19 than those on calcineurin inhibitors or everolimus (mTOR inhibitor). Interestingly, calcineurin inhibitors can combat SARS-CoV-2 viral replication, according to a few studies [88,89]. Therefore, patients with liver transplants receiving mycophenolate drugs might shift to calcineurin inhibitors or everolimus upon encountering COVID-19 infection [90]. However, in patients with severe or rapidly progressive COVID-19, the dose of calcineurin inhibitors should be reduced and antimetabolite drugs should be discontinued while taking antiviral therapy. Although dose adjustment in patients with mild to moderate COVID-19 is generally not recommended. Importantly, patients should be closely monitored [91].

## 6. Drug-Induced Liver Injury during Treatment

For COVID-19 patients with potential drug-induced liver injury (DILI), a dose reduction in drugs should be considered. DILI is defined as liver damage caused by the drug and/or its metabolites or due to its hypersensitivity [92]. Due to the unprecedented pandemic, it was necessary to search for a quick and effective solution through drug repurposing [93]. Kulkarni et al. performed a meta-analysis that showed a pooled incidence of DILI of 25.4% in patients with confirmed SARS-CoV-2 infection [94].

### 6.1. Antiviral Agents

Although using antiviral agents as a treatment of SARS-CoV-2 in patients with chronic viral hepatitis requires close monitoring, it does not strictly preclude the use of antivirals such as remdesivir or immunosuppressors. On the other hand, it was found that the number of patients receiving lopinavir/ritonavir with an abnormal liver function was significantly higher than that of patients with normal liver function [95].

### 6.2. Non-Steroidal Anti-Inflammatory Drugs (NSAIDs)

Theoretically, NSAIDs are rare causes of liver injury, especially upon prolonged use. The apparent mechanism by which almost all NSAIDs induce hepatic injury is idiosyncrasy rather than intrinsic toxicity. Clinically, liver injury from NSAIDs is rare (~1–10 cases per 100,000 prescriptions) and is typically presented as acute hepatitis within 1 to 3 months of starting the medication. Diclofenac and Sulindac are commonly associated with hepatotoxicity; however, all NSAIDs have been reported to cause DILI [96].

However, the multicenter prospective study by Reese et al. showed that NSAIDs use was not associated with higher COVID-19 severity, mortality, invasive ventilation, acute kidney injury, or extracorporeal membrane oxygenation (ECMO) use. They also negated any association between NSAIDs administration and the occurrence of liver injury among 38 centers involved in the study [97]. Similarly, Kushner et al. analyzed 25 studies and revealed that NSAIDs use was not associated with worsened outcomes in COVID-19 patients [98].

### 6.3. Antibiotics

According to the Center of Disease Control and Prevention (CDC), the first year of the pandemic witnessed more than 29,400 deaths from antimicrobial-resistant infections in the USA. Around 40% of those patients caught the infection during hospitalization, with an alarming increase of 15% in resistant infections (https://www.cdc.gov/drugresistance/covid19, accessed on 1 October 2022). Noteworthy, azithromycin, which is the most consistently studied antibiotic for treating SARS-CoV-2, did not improve mortality after 28 days [99]. In addition, it did not affect the clinical course for hospitalized adults with COVID-19. As a macrolide antibiotic, azithromycin administration may rarely lead to an acute, transient, and asymptomatic elevation in serum aminotransferases (1–2% when the drug is used for short periods and a slightly higher percentage with long-term use) [100].

### 6.4. Immunomodulators

The IL-6 antagonist, tocilizumab, is one of the most frequent immunomodulators administered in critically ill COVID-19 patients. However, mild to moderate elevations in transaminases have been observed in COVID-19 patients treated with tocilizumab. This may be explained by the fact that immunomodulators such as tocilizumab, and tofacitinib, can reactivate HBV in patients with occult infection and induce liver damage. Similarly, the corticosteroid frequently used in severe COVID-19 may also cause HBV reactivation [94].

## 7. Conclusions

Liver diseases constitute a burden globally and in the MENA region. Moreover, liver injury was reported during the early era of the pandemic COVID-19 infection. Different degrees of histopathological abnormalities have been observed in the liver tissues of patients with COVID-19 infection, including lobular and sinusoidal lymphocytic infiltration, mild steatosis, and hepatic necrosis. Findings of more significant liver injury include thrombosis of hepatic sinusoids and vessels, whereas bile duct damage is rarely reported. Mechanisms of liver injury are rather complex, involving direct viral attack, potential hepatotoxicity by medications, and COVID-19 hyperinflammatory response. In addition, hypoxia and coagulopathy may underlie liver damage. The deterioration of chronic liver pathology or the occurrence of emerging chronic liver disease may complicate the COVID-19 recovered cases and contribute to long-term COVID-19 effects. The management of COVID-19 infection still remains challenging due to underlying liver diseases as well as risk factors that may affect disease prognosis.

## Figures and Tables

**Figure 1 jcm-12-02067-f001:**
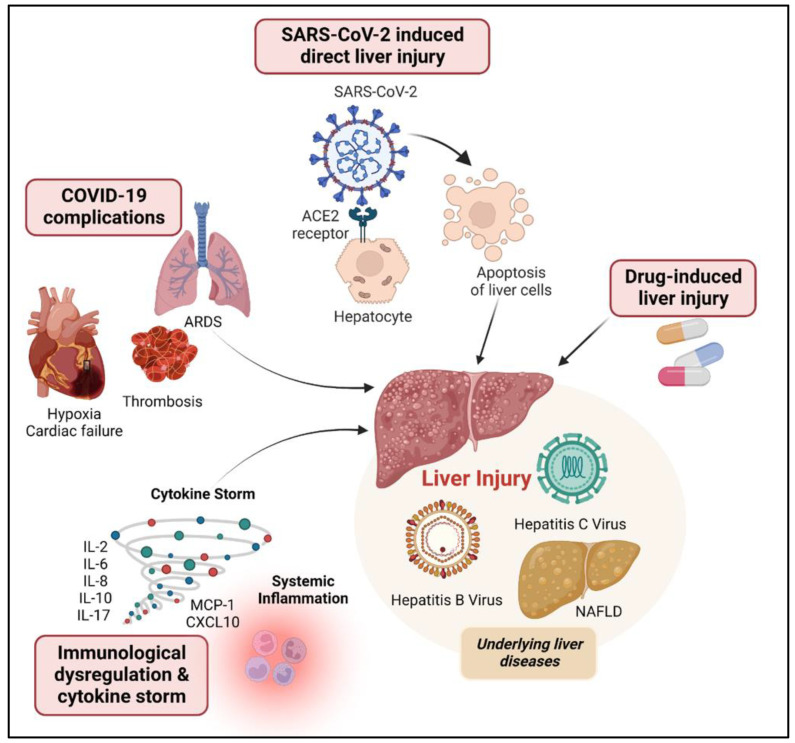
Pathogenesis of liver injury associated with COVID-19 infection. SARS-CoV-2 direct injury, COVID-19 complications, immunological dysregulation, cytokine storm, and drugs are possible causes of liver injury observed in COVID-19 patients. Besides, underlying liver diseases might aggravate liver damage in COVID-19.

## Data Availability

No new data were created or analyzed in this study. Data sharing is not applicable to this article.

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
