# Peer review of "Liver Injury Associated with COVID-19 Infection: Pathogenesis, Histopathology, Prognosis, and Treatment"

_jcm, 2023, doi:10.3390/jcm12052067_

Round 1
Reviewer 1 Report
The manuscript titled “liver injury in covid-19: the status in the middle east” authored by Noha Elemam et al. reviewed the risk factors, and the underlying causes of impaired liver functions in COVID-19. They concluded that different degrees of histopathological abnormalities have been observed in the liver tissues of patients with COVID-19 infection.
My concern is that the paper intended to review the status in the middle east as the title indicated. However, I can not really see that in the paper. The authors reviewed studies related to liver injury and COVID-19 from overall the world. And therefore, I can not see novelty and differences between this review and which has been already published. Authors should include only cases and studies from the middle east and north Africa.
- Authors claimed that the review will have a special focus on the MENA region. However, this is not reflected in the title as the authors said “the status in the middle east”.
- Authors briefly described the background and ended the abstract by highlighting the aim of the review. Also, the authors did not talk about the main finding of the review. I would suggest mentioning the aim of the review earlier in the abstract and then presenting the main findings of the review.
- Figure 1 is not cited in the text.
Although a relatively large effort has been paid to tackle this area of research, the manuscript needs major revision with grammar, spelling, and punctuation corrections.
Author Response
We would like to thank the reviewer for the comments and feedback.
1. We updated section #2 on the epidemiology of cirrhosis and chronic liver disease and added three references (below), to describe the global burden of the disease. The MENA region was discussed in terms of predisposing liver diseases along with the global reports and so the title was amended to fit the contents of the review article.
- Asrani SK, Devarbhavi H, Eaton J, Kamath PS. Burden of liver diseases in the world. Journal of hepatology. 2019;70(1):151-71.
- Jepsen P, Younossi ZM. The global burden of cirrhosis: A review of disability-adjusted life-years lost and unmet needs. Journal of hepatology. 2021;75 Suppl 1:S3-s13.
- Mokdad AA, Lopez AD, Shahraz S, Lozano R, Mokdad AH, Stanaway J, et al. Liver cirrhosis mortality in 187 countries between 1980 and 2010: a systematic analysis. 2014;12(1):1-24.
The aim and findings of the review was clearly stated in the abstract and text of revised manuscript as highlighted in yellow.
“This review discusses the risk factors, and the underlying causes of impaired liver functions in COVID-19, with a focus on key players in the pathogenesis of liver injury. It also highlights the histopathological changes encountered in postmortem liver tissues as well as potential predictors and prognostic factors of such injury, in addition to the management strategies to ameliorate liver damage. A spectrum of liver pathology could be detected in patients with COVID-19 ranging from mild elevation of liver enzymes, to the occurrence of sinusoidal thrombosis. Drug injury is a significant factor complicating the pathophysiology of liver injury and intensifying its impact on COVID prognosis”.
Figure 1 is now added and cited in the text in the revised manuscript.
The manuscript was thoroughly checked and revised for grammar, spelling, and punctuation corrections.
Reviewer 2 Report
The submitted review is comprehensive and exaustive; however, I suggest to shorten the dsetailed description of the study reported and outline only he major findings. This would facilitate the readers and prevent their loosing interest.
Author Response
The authors would like to thank the reviewer for the comments and suggestions. We revised the manuscript and believe that studies were presented appropriately to allow good understanding of the findings. We have thorougly revised the manuscript for spelling errors.
Round 2
Reviewer 2 Report
The manuscript has been modified and the English style has been improved.